# β-Adrenoreceptors as Therapeutic Targets for Ocular Tumors and Other Eye Diseases—Historical Aspects and Nowadays Understanding

**DOI:** 10.3390/ijms24054698

**Published:** 2023-02-28

**Authors:** Elsa Wilma Böhm, Bernhard Stoffelns, Adrian Gericke

**Affiliations:** Department of Ophthalmology, University Medical Center, Johannes Gutenberg University Mainz, Langenbeckstrasse 1, 55131 Mainz, Germany

**Keywords:** β-adrenoreceptors, catecholamines, glaucoma, ocular hemangioma, uveal melanoma

## Abstract

β-adrenoreceptors (ARs) are members of the superfamily of G-protein-coupled receptors (GPCRs), and are activated by catecholamines, such as epinephrine and norepinephrine. Three subtypes of β-ARs (β_1_, β_2_, and β_3_) have been identified with different distributions among ocular tissues. Importantly, β-ARs are an established target in the treatment of glaucoma. Moreover, β-adrenergic signaling has been associated with the development and progression of various tumor types. Hence, β-ARs are a potential therapeutic target for ocular neoplasms, such as ocular hemangioma and uveal melanoma. This review aims to discuss the expression and function of individual β-AR subtypes in ocular structures, as well as their role in the treatment of ocular diseases, including ocular tumors.

## 1. Introduction

Catecholamines, such as adrenaline and noradrenaline, act via α- and β-adrenoceptors (ARs) to regulate central physiological functions, such as blood pressure, heart rate, and contractility, as well as metabolic and central nervous system functions [1]. ARs are members of the superfamily of guanosine triphosphate-binding protein (G-protein)-coupled receptors (GPCRs) [1]. Based on their pharmacological properties, amino acid sequences, and signaling mechanisms, ARs can be divided into three subfamilies: the α_1_-, α_2_-, and the β-AR subfamily [2,3]. The purpose of this review was to give an overview on the expression and function of β-Ars in ocular structures and tumors including those of the periocular region. Moreover, treatment approaches for ocular diseases and tumors targeting β-ARs are presented. The literature was identified via a search on PubMed. The PubMed database search included the following keywords: (“β-adrenoreceptors” OR “β-adrenoreceptor subtypes” OR “β_1_-adrenoreceptors” OR “β_2_-adrenoreceptors” OR “β_3_-adrenoreceptors” OR “β-adrenoreceptor antagonist” OR “β-blocker” OR “catecholamines”), AND (“cornea” OR “conjunctiva” OR “lacrimal gland” OR “trabecular meshwork” OR “uvea” OR “retina” OR “ocular tumors” OR “ocular hemangioma” OR “periocular infantile hemangioma” OR “choroidal hemangioma” OR “retinal hemangioblastoma” OR “conjunctival hemangioma” OR “uveal melanoma”). The research was performed from 13 November 2022 to 23 January 2023 with the following inclusion criteria: all studies, written in English, and published after 1948. Studies reporting on the role of β-adrenoceptors in non-(peri)ocular tissues were excluded. The reference list of all selected articles was reviewed for further identification of potentially relevant studies.

## 2. Historical Aspects, Classification and Function of β-Adrenoceptors

The theory on the existence of specific receptors binding drugs or intracellular molecules for regulation of cellular mechanisms was first introduced by John Newport Langley in the early 1900s [4]. The hypothesis, that these receptors are specific and selective, was proposed by Paul Ehrlich at the same time [5]. The first description of α- and β-adrenoceptors activated by epinephrine was published in 1948 by Raymond P. Ahlquist [1]. In 1958, the first β-blocker was synthesized by Eli Lilly Laboratories [4]. A few years later, Sir James W. Black introduced the first clinical application of β-blockers treating angina pectoris with propranolol, receiving the Nobel Prize for this crucial success in clinical medicine [4]. Over time, several clinical trials, especially in cardiovascular research, were performed, and β-Ars and their antagonists became a central tool in the treatment of cardiovascular diseases, such as angina pectoris, hypertension, arrhythmias, and post myocardial infarct [4].

Binding of an agonist to a β-adrenoceptor causes a dissociation of Gα-GTP and Gβγ subunits with consequent activation of adenylate cyclase, and production of the second messenger cyclic adenosine monophosphate (cAMP). This leads to activation of downstream effectors, such as cAMP-dependent protein kinase (PKA) and cAMP-dependent phosphorylation of gated ion channels [6,7]. Among the β-AR family, three different subtypes (β_1_-, β_2_-, and β_3_-ARs) can be distinguished based on pharmacological studies [8]. The previously postulated β_4_-AR turned out to be a specific affinity state of the β_1_-AR [9]. The β_1_-AR has high affinity to adrenaline and noradrenaline, and is expressed predominantly in the heart, brain, and adipose tissue. A lower affinity to noradrenaline is found in the β_2_-AR subtype, which is more involved in relaxation of vascular and other smooth muscle cells, and further metabolic mechanisms of catecholamines [10]. The β_3_-AR is abundantly expressed in adipose tissue, but is also expressed in the eye [11,12]. β-ARs are found in the central and peripheral nervous systems, and are essential for a variety of functions, such as regulation of heart rate and contractility, vasorelaxation, bronchodilation, and neurotransmitter release [13]. In this review, we focus on the expression and function of β-ARs in ocular structures. Furthermore, we describe the role of these receptors in ocular diseases, including ocular tumors, such as hemangioma and uveal melanoma. The role of β-blockers, such as propranolol, as possible therapeutic tools will also be discussed. A scheme of β-adrenergic the signaling pathways is shown in Figure 1.

## 3. β-Adrenoceptors in Ocular Structures

### 3.1. Cornea

Adrenergic nerve fibers originating from the superior cervical ganglion innervate the surface of the cornea. In adult human corneas, these nerve fibers could be identified by sodium-potassium-glyoxylic acid-induced fluorescence [14,15]. Both α-ARs and β-ARs have been detected in corneas, and the mediating neurotransmitter, norepinephrine, has been identified in the corneal epithelium [15,16,17,18,19]. Furthermore, high levels of β_2_-ARs have been detected in corneal epithelial cells [20,21,22]. There is an ongoing discussion on the role of β_2_-ARs in corneal re-epithelialization [23,24,25]. Some researchers suggested that, through antagonism of β_2_-ARs, and consequent increase in extracellular signal-regulated kinase (ERK) phosphorylation, corneal epithelial cell migration and corneal wound healing was enhanced [22,26]. Recently, a study was published revealing attenuated corneal wound healing after treatment with β_2_-AR antagonists by modulating the expression of Ki67, and phosphorylation of ERK1/2 in the limbal and regenerated corneal epithelium [27]. These contrary results point toward a role of the β_2_-AR in homeostasis of the corneal epithelium. However, the definite role of β-AR signaling in corneal wound healing remains to be examined in further studies.

### 3.2. Conjunctiva

Conjunctival functions, such as secretion of mucous substances from goblet cells, and mobilization and attraction of conjunctival eosinophils, seem to be controlled by sympathetic nerves. These functions might be involved in the pathophysiology of common conjunctival diseases, such as allergic conjunctivitis and dry eye disease [28,29,30,31].

In cell culture experiments with primary human conjunctival epithelial cells, the presence of β_2_-ARs was also demonstrated [32]. Using fluorescence microscopy, β_1_- and β_2_-ARs could be detected in conjunctival goblet cells in developing rats [33]. The β_3_-AR subtype was only seen on epithelial and goblet cells of the human conjunctiva, but not in conjunctival cells of the mouse [28]. In the human conjunctival epithelial cell line, IOBA-NHC, all individual AR subtypes could be detected by western blot analysis. In the same study, analysis by flow cytometry revealed constitutive expression of β_1_- and β_3_-ARs on cell membranes and in intracellular compartments. The β_2_-AR was detected only intracellularly under normal culturing conditions. Furthermore, immunofluorescence microscopy was performed, and β_1_- and β_2_-AR subtypes, but not β_3_-ARs, were detected in IOBA-NHC cells [34]. In biopsies of the human conjunctiva, all β-AR subtypes could be detected [34]. Notably, β-AR subtypes seem to play a role in the pathogenesis of certain conjunctival diseases. For example, irregular expression of the β_1_-AR was observed in all epithelial cell layers of human conjunctival biopsy specimens of patients with vernal keratoconjunctivits, suggesting an involvement of the autonomic nervous system in pathogenesis of the disease [35]. Furthermore, β_2_-AR agonists, such as salbutamol and terbutaline, reduced microvascular permeability, and exerted anti-inflammatory effects in allergic conjunctivitis [29,36]. These findings reveal that β-ARs might be a promising therapeutic target for inflammatory ocular surface diseases, such as allergic conjunctivitis.

### 3.3. Lacrimal Gland

The lacrimal gland is composed of acinar cells, myoepithelial cells, and ductal cells [37]. With pharmacological tools, detection of β-ARs in the lacrimal gland of various species was possible [38]. By using RT-PCR, mRNA for all AR subtypes except for α_2C_-, β_1_-, and β_3_-ARs was detected in acinar cells of the rat lacrimal gland [39]. However, protein secretion in the lacrimal gland of rats and mice was reported to be regulated by stimulation of β-ARs [40,41,42,43]. Moreover, in rabbits, contribution of β_1_- and β_2_-ARs to secretion in the lacrimal gland was suggested [42,44]. Nevertheless, the physiological and pathophysiological roles of ARs in accessory lacrimal glands need to be discussed. Immunohistochemical studies in human specimens indicated that β_1_-ARs were the predominant AR subtype in the glands of Wolfring [45]. In epithelial cells of the human meibomian gland, activation of β_2_-ARs was suggested to stimulate lipid synthesis [46]. These findings indicate that β-ARs are participating in regulation of tear secretion. They may also play a crucial role in the pathophysiology of dry eye disease, including Sjögren’s syndrome, by an altered neuronal control of lacrimal gland fluid regulation [37,39]. Based on these findings, activation of adrenergic pathways may be a potential therapeutic approach to treat dry eye disease.

### 3.4. Trabecular Meshwork

In sections of the trabecular meshwork, predominant expression of β_2_-ARs has been shown [47]. Moreover, high expression and functional relevance of β_2_-ARs was detected in pharmacological and radioligand binding studies in cultured human trabecular cells and in trabecular meshwork from human eyes [48,49,50]. An augmentation of outflow facility through the trabecular meshwork mediated via ß-AR signaling with a particular role of the β_2_-AR after adrenaline and noradrenaline stimulation was demonstrated in studies on monkey and human eyes [51]. Further functional studies on isolated trabecular meshwork strips revealed that β-adrenergic agonist stimulation induced relaxation, whereas contraction was induced by α-adrenergic agonists [52]. In summary, there is some evidence that activation of β_2_-ARs increases outflow facility in the trabecular meshwork.

### 3.5. Uvea

The uvea consists of three parts: the iris, the ciliary body, and the choroid. In the human ciliary body, abundant expression of β_1_- and β_2_-ARs was reported [53,54,55,56,57]. Furthermore, autoradiographic and ligand binding studies in rabbit eyes revealed expression of β-ARs in the ciliary process epithelium, indicating that they may participate in aqueous humor formation [58,59]. β_2_-ARs coupled to adenylate cyclase were detected in the ciliary process epithelium in various species, including humans [60,61,62]. Adrenergic agonists, such as adrenaline, are suggested to induce a desensitization of the β-AR-adenylate cyclase complex, which may be a reason for the delayed intraocular pressure decrease after topical application of adrenergic agonists, and for the paradoxical fact that both β-AR agonists and antagonists lower intraocular pressure [63]. Vasoconstriction of the uveal vasculature with consequent decrease in aqueous humor production could be another possible mechanism of the intraocular pressure-lowering effect of β-AR blockers [64]. Due to potent binding of the non-subtype-selective β-AR antagonist, timolol, and some β_2_-AR antagonists to β-ARs on ciliary processes in radioligand binding studies, and potent lowering of intraocular pressure in various species, β_2_-ARs are a pharmacological target in glaucoma treatment [65,66,67,68]. Abundant expression of β-ARs has also been detected in the choroid [58]. In humans, the presence of β-ARs in the choroid was found by showing an increased choroidal vascular tone following systemic administration of the nonselective β-AR blocker, timolol [69]. Sympathetic nerves and signaling via β-ARs may be important for maintaining normal choroidal vascular architecture [70]. Experiments with the specific β_3_-AR agonist, BRL37344, revealed that the β_3_-AR may be involved in choroidal endothelial cell invasion and elongation [12]. Moreover, the β_2_-AR may participate in the regulatory process of VEGF and IL-6 expression in cells of the choroidal endothelium and other cells, indicating that blockade of these receptors may attenuate formation of choroidal neovascularization [71,72].

### 3.6. Retina

#### 3.6.1. Potential Sources of Catecholamines in the Retina

The retina is a complex neuronal multilayer processing visual information to the brain [73]. Six major types of neuronal cells are found in the neuronal lamination of the retina: retinal ganglion cells (RGCs), amacrine cells, bipolar cells, horizontal cells, and the cone and rod photoreceptors [74]. These retinal neurons use same types of neurotransmitters (noradrenaline, dopamine, and acetylcholine) as those of the central nervous system [75]. Due to the complex neuronal structure of the retina, it is vulnerable to various detrimental factors, such as ischemia and oxidative stress, that may lead to deterioration of retinal cell function, and consequently cause retinal pathologies [76].

The supply of the retina with oxygen and other nutrients is ensured by two different vascular beds, both originating from the ophthalmic artery, the retinal circulation, and the choroidal circulation [77]. Choroidal blood vessels are innervated and modulated by autonomic nerve fibers. In contrast, such nerve fiber terminals were not found in or on the wall of human retinal blood vessels [78]. Vascular tone of retinal blood vessels is controlled by local chemical factors, including oxygen (O_2_), carbon dioxide (CO_2_), nitric oxide (NO), and hydrogen sulfide (H_2_S) [79,80,81]. Although there is no evidence for sympathetic nerve fibers in the retina, catecholamines including noradrenaline, adrenaline, and dopamine have been detected in retinal tissue [82]. A potential source of noradrenaline in the mammalian retina could be sympathetic nerve terminals located in the choroid, reaching ARs of the retina by paracrine diffusion [83,84]. In support of this concept, decreased retinal noradrenaline concentrations have been reported after removal of the superior cervical ganglion, that provides sympathetic input to the choroid [82].

The predominant catecholamine in the retina is dopamine, synthetized and released by dopaminergic amacrine (DA) cells [85,86]. A potential source of noradrenaline could be metabolization of dopamine to noradrenaline in retinal tissue [87]. Apart from noradrenaline, dopamine may also activate α_1_-, α_2_-, and β-ARs [88]. It is important to recognize that members of all three adrenoceptor subfamilies, α_1_-, α_2_-, and β-ARs, have been detected in retinal tissue, including endothelial cells of retinal blood vessels [89,90,91,92,93,94].

Sympathetic neurotransmission presents a possible mechanism to regulate expression of inducible nitric oxide synthase, angiogenic growth factors, and the number of pericytes in the retina, as described below [95,96,97].

#### 3.6.2. Expression of β-Adrenoreceptors in the Retina

β-ARs were detected in the outer nuclear layer, the outer plexiform layer, the inner nuclear layer, and the inner plexiform layer of the rat retina [98]. In the human retina, β-ARs were visualized in vitro by autoradiography [21].

In bovine retinal vessels and in the neural retina, β_1_-AR and β_2_-AR binding sites have been detected [99]. Functional β_3_-ARs have also been found in rat retinal blood vessels [100]. Immunohistochemical studies revealed localization of β_3_-ARs in the inner capillary plexus of the mouse mid-peripheral retina [101,102]. Intriguingly, pharmacological activation of β_3_-ARs induced cell migration and proliferation of retinal vascular endothelial cells [93]. Furthermore, expression of β_1_- and β_3_-ARs has been reported in human retinal endothelial cells [103]. Due to the broad distribution of β-ARs in retinal blood vessels and in the neural retina, β-ARs are believed to play an important role in the regulation of vascular and neuronal functions of the retina.

#### 3.6.3. Role of β-Adrenoreceptors in the Retina

During stress conditions, such as hypoxia, catecholaminergic responses from the cardiovascular system are enhanced leading to activation of β-ARs [104]. Under hypoxic conditions, increased levels of noradrenaline by approximately 90% were detected in the mouse retina compared to normoxic conditions [102]. Activation of β-ARs induces an upregulation of hypoxia-inducible factor-1α (HIF-1α) and vascular endothelial growth factor (VEGF). These factors maintain a crucial role in the formation of pathogenic blood vessels in various retinal diseases, such as ROP and diabetic retinopathy [102,105,106]. By application of propranolol, a non-subtype-selective β-AR antagonist, this hypoxia-mediated increase in VEGF expression, which is involved in neovascularization of the retina, could be prevented [84]. Likewise, in a mouse model of oxygen-induced retinopathy (OIR), subcutaneous administration of propranolol resulted in a decrease in VEGF and HIF-1α levels, indicating that blockade of β_1_- and β_2_-ARs is protective against retinal angiogenesis, and can enhance blood-retinal barrier function [101]. Another study that used a mouse model of OIR also revealed reduced retinal VEGF receptor-2 expression, and less vascular abnormalities in the superficial plexus of the retina after deletion of the β_1_- and β_2_-ARs [107]. Selective blockade of β_2_-AR by ICI 118,551 in a mouse OIR model also resulted in decreased retinal levels of proangiogenic factors, and reduced pathogenic neovascularization, suggesting that β_2_-AR blockade may be a potential way to block retinal angiogenesis [108]. However, there are also contradictory findings that need to be mentioned. While most studies reported inhibitory effects of pharmacological β-AR blockade on VEGF levels, some other studies observed a blockade of VEGF formation by exposure to β-AR agonists [84,101,102,104,105,106]. For example, decreased VEGF levels in the diabetic rat retina were reported after administration of a novel β-AR agonist, compound 49b [109]. Modulation of eNOS and PKC pathways with an increase in insulin-like growth factor binding protein 3 (IGFBP-3) may be the reason for reduced VEGF levels in the diabetic retina [109].

These contradictory findings lead to the suggestion that the effects may be mediated through diverse regulatory mechanisms depending on the retinal disease and the experimental setting.

Previous studies reported that the nonselective β-ARs agonist, isoproterenol, can cause agonist-induced β_2_-AR desensitization that downregulates expression of β_2_-ARs in the retina, which in turn exerts an inhibitory effect on VEGF expression in OIR [102].

β_3_-ARs were shown to have an impact on neovascularization processes of various retinal vascular diseases [101]. For example, in an OIR mouse model with dense β_3_-AR immunoreactivity in engorged retinal tufts, upregulation of β_3_-ARs in response to hypoxia indicated that activation of β_3_-ARs also plays an important role in pathologic angiogenesis [101]. Furthermore, hypoxia-inducible factor-1α (HIF-1α) could increase the expression of the β_3_-AR gene in the hypoxic retina, supporting the hypothesis that β_3_-ARs may participate in the angiogenic response in hypoxia [110]. Due to severe side effects, such as thromboembolism or hyperkalemia, systemic administration of HIF-1α inhibitors is not recommended [111,112]. Downregulation of retinal VEGF release via modulation of the nitric NO signaling pathway could be induced by the β_3_-AR antagonists, L-748,337, and SR59230A [113]. Furthermore, retinal damage after intravitreal injection of N-methyl-D-aspartate (NMDA) in rats could be prevented by the β_3_-AR agonist, CL316243 [114]. In addition, β_3_-ARs, which are not coupled to a G protein receptor kinase are resistant to agonist-induced desensitization [115,116]. Therefore, the β_3_-AR may represent an attractive therapeutic target for the treatment of ischemic retinal diseases.

β-AR stimulation was also shown to increase human and mouse pericyte survival under diabetic conditions [117]. In support of this concept, surgical removal of the right superior cervical ganglion, which supplies the eye with sympathetic nerve fibers, induced a significant decrease in the number of retinal pericytes in a rat model [97]. These findings indicate that β-AR signaling is important for pericyte survival.

Moreover, β-ARs are involved in regulation of inducible nitric oxide synthase (iNOS) expression [118]. Activation of β-ARs reduced levels of iNOS and other inflammatory molecules, such as interleukin (IL)-1β, tumor necrosis factor-α (TNF-α), and prostaglandin E2 (PGE2) in human retinal endothelial cells, and rat Müller cells in an in vitro model of hyperglycemia [95]. In line with these findings, iNOS expression in human retinal endothelial cells grown in high glucose medium could be reduced by the partial β_1_-AR agonist xamoterol [95]. An overview on the expression and function of β-ARs in ocular structures is presented in Table 1.

## 4. β-Adrenoreceptors in Ocular Tumors

### 4.1. Periocular Infantile Hemangioma

Infantile hemangioma is a benign vascular tumor, which typically occurs as a cutaneous lesion in early childhood with a high incidence of approximately 4–5% of infants [119,120]. It is characterized by nonlinear growth during its proliferative stage, reaching its final size by the age of about 3 months [121]. Periocular infantile hemangioma can present as a small isolated lesion or as a large mass with visual impairment through ocular occlusion [120]. A major part of the tumors is characterized by a predictable course with spontaneous regression. However, persisting lesions can lead to serious ocular or systemic complications, such as amblyopia or cardiac failure [120]. There are different growth patterns in infantile hemangioma. Superficial lesions present as red lesions with a flat or rough surface. Deep hemangiomas grow later and longer than superficial hemangiomas, and appear as blue to purple discolorations or changes of thickness of the skin. They can also only cause anatomical distortions without discoloration [120,121]. Infantile hemangioma is frequently located unilaterally at the upper eyelid [122]. Twenty to 40% of large facial segmental hemangiomas are associated with the PHACE (posterior fossa anomalies, arterial, cardiac, eye, and endocrine anomalies) syndrome related to typical malformations, such as hemangioma, posterior fossa abnormalities present at birth, arterial lesions, cardiac anomalies, eye abnormalities, and sternal clefting [123,124]. These clinical manifestations can lead to systemic, cutaneous, and ocular complications. Periocular manifestation of infantile hemangioma can cause severe ocular complications, such as ptosis, strabismus, telangiectasia, ulceration, scarring, or facial disfigurement [120]. Secondary vision loss due to amblyopia occurs in 60% and represents the most common ocular complication in patients with periorbital hemangioma [125]. Systemic complications include airway obstructions or high cardiac output problems [120]. Following clinical examination, imaging techniques such as ultrasonography, color Doppler ultrasonography [126], computer tomography scans, and magnetic resonance imaging and angiography play a central role in the diagnosis of hemangiomas [120,127,128]. The main therapeutic target in the treatment of infantile periocular hemangioma is the prevention of systemic and ocular complications, such as amblyopia. Due to a high spontaneous remission rate, small lesions without risk of clinical complications have to be observed in first line [120]. There are medical and surgical treatment modalities described in infantile hemangiomas. Medical treatment strategies next to beta blockers include systemic or intralesional corticosteroids, imiquimod, vincristine, bleomycin A5, cyclophosphamide, interferon-α, or angiotensin-converting enzyme (ACE) inhibitors, such as captopril [120,129]. Less common modalities are laser therapy or surgery [129].

The therapeutic effect of β-blockers on infantile hemangioma was found by chance, when patients were treated with β-blockers for cardiopulmonary indications, and regression of hemangioma was observed [130]. Since then, several studies on the application of the nonselective β-blocker, propranolol, in infantile hemangioma have been published. Hermans et al. performed a prospective study in 174 patients with complicated hemangioma, and reported treatment success in 99.4%. Treatment success was defined as immediate cessation of growth, softening, fading of the erythema, and rapid induction of regression [131]. With a fixed dose of 2 mg/kg bodyweight oral propranolol, complete regression of hemangioma occurred in 60% of the treated patients. Among the patients, 17.4% showed evidence of rebound growth after termination of therapy, but they responded well to re-treatment [132]. The respective study of Schneider et al. included 207 patients and also found high effectivity of systemic application of propranolol [133]. Potential side effects were arterial hypotension (3.4%), wheezing (9.2%), nocturnal restlessness (22.4%), and cold extremities (36.2%) [131]. Others reported severe hypoglycemia during treatment with propranolol [134]. This side effect was not observed when propranolol was strictly applicated after feeding [133,135]. Furthermore, side effects, such as bronchial hyperreactivity or constipation, have been observed [136]. However, it is reasonable to start therapy with propranolol under severe surveillance. A multicenter retrospective analysis demonstrated superiority of oral propranolol against oral corticosteroids, with respect to clinical and cost efficacy. Furthermore, fewer surgical interventions and fewer adverse effects were found in the propranolol treated group [137]. To minimalize adverse effects, topical application of β-blockers, such as timolol, can also be performed to treat infantile periocular hemangioma. Especially in localized and superficial hemangiomas, successful treatment with topical timolol maleate 0.5% gel has been reported [138,139,140,141,142]. Chambers et al. also demonstrated good results of topical timolol in a retrospective study [143]. Likewise, studies showing good response of deep hemangiomas to topical timolol application have been published [144,145]. The timolol maleate 0.5% solution or gel should be applied twice daily on the surface of the lesion, and therapeutic responses can be seen after 4–8 weeks [120]. On the other hand, a randomized clinical study found limited benefit of topical timolol in lesion resolution when given during the early proliferative stage compared to a placebo treated group [146].

A randomized controlled trial found that combination of systemic and topical β-blockers in the early proliferative stage is significantly more effective than systemic treatment only [147]. There are also few investigations analyzing intralesional β-blocker application. For example, Awadein et al. showed a reduction in hemangioma lesion size, asigmatic error, and degree of ptosis after intralesional propranolol injection. There were no adverse effects reported. The rate of rebound growth was 25%, but with good response to reinjection [148]. Others reported interruption of hemangioma growth without changes in size or color after intralesional propranolol injection [149]. Another study compared intralesional and oral propranolol application, and found comparable results in efficacy and side effects [150]. In summary, β-blockers represent an effective and safe possibility to treat infantile periorbital hemangioma.

### 4.2. Choroidal Hemangioma

Choroidal hemangioma is a benign vascular tumor located at the ocular posterior pole. It can be distinguished between circumscribed choroidal hemangioma (CCH) and diffuse choroidal hemangioma. Circumscribed choroidal hemangioma emerge sporadically and are not associated to local or systemic abnormalities. CCHs are solitary, well-demarcated lesions located posterior to the equator [151]. They are mostly detected in the second to fourth decade of life when visual restrictions occur. Diffuse choroidal hemangiomas are usually part of neuro-oculo-cutaneous hemangiomatosis (Sturge-Weber syndrome) and manifest at birth [152]. Visual symptoms occur due to subretinal fluid, cystoid macular edema, changes in retinal pigment epithelium, subretinal fibrosis, retinoschisis, or exsudative retinal detachment [151]. Common symptoms are decreased visual acuity (81%), visual field defects (7%), metamorphopsia (3%), floaters (2%), progressive hypermetropia (1%), photopsia (1%), pain (1%), and no symptoms (6%) [153]. At the ocular fundus, a circumscribed orange-red tumor can be seen, that is found to be unilateral and located most frequently in the superotemporal quadrant close to the macula [154]. Few cases of bilateral CCH have also been reported [155,156,157]. Diagnostic tools next to fundus examination are ultrasound, fluorescent angiography, indocyanin green (ICG) angiography, enhanced-depth imaging (EDI) optical coherence tomography (OCT), and OCT-angiography (OCT-A) [151]. Representative pictures of a patient with circumscribed hemangioma are shown in Figure 2. Treatment of CCH is indicated when clinical symptoms occur. The primary goal of therapeutic strategies is reduction in subretinal fluid and macular edema, which causes a decrease in visual acuity. Reduction in tumor size is only an additional outcome [151]. Therapeutic strategies are laser photocoagulation [158], photodynamic therapy (PDT) [159], transpupillary thermotherapy (TTT) [160], and radiation therapy, such as external beam radiotherapy (EBRT) or episcleral brachytherapy [161,162]. These therapies have often been combined with intravitreal anti-VEGF injections [163,164].

The pathophysiology of choroidal hemangioma Is not fully understood. Choroidal hemangioma is supposed to be a congenital vascular hamartoma that is composed of choroidal vessels. Histological studies revealed that choroidal hemangiomas are nonproliferative tumors with tumor growth by venous congestion rather than cell proliferation [165,166]. In cutaneous capillary hemangioma, activation of endothelial cells and β-ARs are supposed to be involved in the pathogenesis [167,168] However, some studies using nonselective β-blockers as therapeutic strategy in choroidal hemangioma have also been described so far. Sanz-Marco et al. treated a patient with CCH, who was resistant to treatment with laser photocoagulation, by oral administration of propranolol, and reported improvement of visual acuity, and resolved serous macular detachment without complications through systemic or local adverse effects [169]. Likewise, there have been several case reports demonstrating successful treatment of choroidal hemangioma by oral application of propranolol [170,171,172,173,174]. Reported adverse effects are decreased exercise tolerance, an increase in fatigability, and weakness. A prospective, longitudinal, interventional study treating patients with CCH with oral propranolol at a dosage of 1.5 mg/kg/day found a reduction in sub- and intraretinal fluid without full regression in the first four months of treatment. Afterwards, stagnation in the therapeutic response followed by worsening despite continued therapy occurred, indicating that a saturation point exists [175]. Furthermore, in a case report on oral propranolol treatment in two patients with Sturge-Weber syndrome, no therapeutic effect could be found [176]. Histopathologically, cavernous and capillary components have been found [165]. In patients with Sturge-Weber syndrome, the predominant hemangioma type is the cavernous type, followed by mixed cavernous and capillary types [177]. This could be an explanation for the different therapeutic responses to propranolol in patients with Sturge-Weber syndrome compared to capillary hemangioma [165]. Recently, intravitreal β-blocker application has also been studied. Jorge et al. reported on a single case with CCH and consequent extensive subretinal fluid resistant to intravitreal anti-VEGF therapy. After two following intravitreal injections of metoprolol, a decrease in subretinal fluid, and improvement of visual acuity could be recognized [178]. In a phase I clinical trial, safety and early outcome of intravitreal metoprolol injection was analyzed, and no signs of acute intraocular toxicity were found. Furthermore, the authors also found a decrease in intra- or subretinal fluid [179]. Application of β-blockers intravitreally was also found to be safe in patients with central serous chorioretinopathy, and in rabbits [180,181]. However, further investigations addressing long-term effects and different drug concentrations are mandatory in this field.

### 4.3. Retinal Hemangioblastoma

Retinal hemangioblastoma (RH) is a main component of ocular manifestation of Von-Hippel Lindau (VHL) disease. A national study with 64 genetically tested participants with RH revealed VHL as the underlying cause in 84% of the cases [182]. VHL disease is an autosomal dominantly inherited mutation in the VHL tumor suppressor gene that is related to characteristic benign and malignant neoplasms, such as central nervous system hemangioblastoma, RH, pheochromocytoma, or clear cell renal carcinoma [183]. First manifestation of RH mostly occurs at young age between 25 and 37 years [182,184]. Unilateral location was found in 42.1%, and bilateral location in 57.9% of patients [185]. The main part of RHs is located in the peripheral retina, and is less common in the juxtapapillary area [185]. Clinical manifestations can be vision loss due to retinal exudation, fibrosis, vitreous and subretinal hemorrhage, or exsudative retinal detachment [183]. To confirm the diagnosis of RH, fundus examination, fluorescein angiography, and optical coherence tomography play a central role. Furthermore, genetic testing is important to secure the diagnosis of VHL disease [183]. When RH is detected, fast intervention or close follow-up is needed. There are different established methods for ablative treatment, such as laser photocoagulation, cryotherapy, radiotherapy (brachytherapy, external beam radiotherapy, and proton beam radiotherapy), photodynamic therapy, or transpupillary thermotherapy [183]. Tumor-associated exudation can be reduced by intravitreal injection of VEGF blockers [183,186].

Histological studies revealed that RH are composed of abnormal capillary-like fenestrated channels surrounded by vacuolated stromal cells, and tumorlet-like cells expressing stem cell markers [187]. A loss of heterozygosity within the VHL tumor suppressor gene was mostly found in cells from the hematopoetic/vascular lineage, and was associated with an increase in VEGF, hypoxia induced factor (HIF), or ubiquitin [188]. By the absence of VHL protein, HIF accumulates because it is not ubiquitinated for degradation, and activates the expression of its targeted genes, such as VEGF [189]. Cell culture experiments could show that propranolol induced apoptosis in hemangioblastoma cells from VHL patients by downregulating HIF-dependent transcription [189]. Cuesta et al. could also demonstrate a reduction in activation of HIF-target genes by the β_2_-AR antagonist, ICI-118,551, in hemangioblastomas in VHL disease [190]. Hence, propranolol could be a therapeutic tool to control hemangioblastoma growth in patients with VHL. In a study with oral application of 120 mg propranolol once daily in VHL patients with RH, tumor size remained stable, and retinal exudation decreased without any adverse events. Furthermore, biomarkers, such as VEGF and miRNA, decreased in the first month of treatment [191]. Moreover, other authors reported that hemangioblastomas remained stable during 12 months under therapy with propranolol [192]. These studies indicate that systemic β-blockers are a feasible possibility for treatment of retinal hemangioblastomas, especially when juxtapapillary location complicates ablative treatment.

### 4.4. Conjunctival Hemangioma

Conjunctival hemangiomas are benign vascular tumors located at the bulbar conjunctiva that can grow sessile or pedunculated [193]. Congenitall hemangiomas have already reached their full size at birth. Infantile hemangiomas manifest at birth, but can increase in size followed by involution. In adults, acquired hemangiomas are described, also with increasing size [193,194]. Clinically, they can become apparent by elevated red lesions on the bulbar conjunctiva [195] and possible conjunctival bleeding [193]. For clinical management, regular observation is necessary. Especially in elderly patients, malignancy needs to be excluded. Therefore, biopsy and histopathological examination are mandatory [193].

Conjunctival hemangiomas consist of vascular channels lined by endothelial cells with positive immunostaining signals for CD31 and CD34, but with negative signals for the smooth muscle marker desmin [196]. There are several studies that show promising results for the therapy of conjunctival hemangioma by topical application of non-subtype-selective β-blockers, such as timolol. Lubahn et al. reported that an acquired sessile hemangioma resolved following topical application of timolol for 6 months [197]. In addition, infantile conjunctival hemangiomas were successfully treated by topical timolol administration [198,199].

### 4.5. Potential Mechanisms of Ocular Hemangioma Treatment with Non-Subtype-Selective β-Blockers

As described above, non-subtype-selective β-blockers represent an effective method to treat different types of ocular hemangioma, and these effects have been discovered by chance. Expression of all three subtypes of β-ARs in hemangiomas has been demonstrated [200]. However, the specific role of each subtype is still not fully understood. Recently, it has been shown, that overexpression of the β_3_-AR is associated with a lack of response to propranolol [201].

Early effects of propranolol may be due to intralesional vasoconstriction caused by decreased release of nitric oxide [202]. Catecholamines activate endothelial β_2_-ARs, and induce vasodilatation via endothelial vasorelaxing factors, such as nitric oxide [203]. Through non-subtype-selective blockade of β-ARs, vasoconstriction and consequent reduction in intralesional blood flow can be induced [204,205]. Clinically, these effects cause reduction in the surface redness, brightening, and softening in infantile hemangioma [120,205]. However, longtime effects of propranolol treatment may not be caused by intralesional vasoconstriction.

Intermediate effects of propranolol treatment could be explained by a reduction in proangiogenic factors, such as vascular endothelial growth factor (VEGF), matrix metalloproteinases (MMPs) or proangiogenic cytokines, such as interleukin-6 (IL-6) [205]. Catecholamines upregulate VEGF and HIF alpha-protein through β-ARs by induction of protein kinase A (PKA) and cyclic adenosine monophosphate (cAMP) [206]. It has been reported that chronic behavioral stress increased tumor angiogenesis and growth in ovarian carcinoma cells by β-AR-mediated activation of the cAMP-PKA signaling pathway with consecutive upregulation of VEGF, MMP2, and MMP9 [207]. Prevention of catecholamine stimulation by propranolol was shown to reduce expression of VEGF-A, which is known to play a crucial role in hemangioma growth [205]. By suppression of VEGF-A and VEGF-C, inhibitory effects on the development of experimental hemangioma have been shown [208]. In addition, other proangiogenic factors are downregulated by propranolol. By AR blockade through propranolol in vitro, a reduced tubulogenesis in human brain endothelial cells, and a decreased MMP-9 secretion was observed [209]. For angiogenesis, apart from formation of new blood vessels, different other events are necessary [205]. To enable migration of endothelial cells, proteolysis of components of the extracellular matrix is essential [205]. In this process of carcinogenesis, MMPs play a central role [210]. It has been shown that inhibition of MMPs reduced in vivo hemangioma growth [211]. Moreover, it has been observed that norepinephrine promoted the invasiveness of pancreatic cancer cells, which was associated with an increased expression of MMP-2, MMP-9, and VEGF. Intriguingly, these effects were blocked by propranolol [212]. Another proangiogenic cytokine, whose expression can be induced by catecholamines, is IL-6 [213]. High levels of IL-6 have also been found in infantile hemangioma, and inhibition of IL-6 was shown to reduce hemangioma growth [214]. Propranolol was also found to reduce IL-6 levels by blocking its upregulation [205]. These mechanisms may be responsible for interruption of hemangioma growth during propranolol application.

Long-term effects of β-blockers in hemangiomas may be explained by induction of apoptosis in endothelial cells during the proliferative stage [202,215,216]. Possible mechanisms are activation of pro-apoptotic genes, such as caspase-9, caspase-3, p53, and Bax, and down-regulation of anti-apoptotic genes, such as Bcl-xL [217,218]. Furthermore, propranolol may reduce differentiation of hemangioma progenitor cells into endothelial cells or pericytes [205]. Conversion from a proliferating to an involuting tumor stage could be caused by differentiation of progenitor cells to adipocytes, induced by propranolol [205]. A scheme of the early, intermediate and longtime effects of β-blockers on hemangiomas is shown in Figure 3.

### 4.6. Uveal Melanoma

Uveal melanoma is the most common primary intraocular malignant tumor in adults. In Europe, standardized incidence rates of 1.3–8.6 cases per million per year were reported [219]. However, uveal melanoma is a relatively rare cancer, occurring commonly in older age groups [220]. In Europe and in the United States, the median age of uveal melanoma diagnosis is 59 to 62 years [221]. Epidemiological studies revealed a higher incidence of uveal melanoma in males than in females [222]. The majority of uveal melanomas is located in the choroid (90%). A minor part was found to be located in the iris (4%) and in the ciliary body (6%) [223]. Predisposing factors are presence of choroidal nevus, oculodermal melanocytosis, fair skin, blond hair, light eye color, inability to tan, and mutation of BRCA1-associated protein 1 [220,224]. Furthermore, environmental factors, such as exposure to sunlight or artificial ultraviolet light may play a role in the development of uveal melanoma [220]. Iris melanoma is commonly detected by changes in iris color or pupil distortion. Additionally, it can cause secondary glaucoma by compression of the anterior chamber angle, angle seeding, ectropion uveae, hyphema, and extraocular extension [225]. Ciliary body and choroidal melanoma are characterized by painless vision loss or metamorphopsia due to serous retinal detachment in larger tumors. Other clinical manifestations are blurred vision, photopsia, floaters, visual field loss, visible tumor, or pain [220]. Thirty percent of uveal melanomas remain asymptomatic [220]. About 50% of patients with uveal melanoma develop metastatic disease, most frequently located in the liver, which has a high impact on the patient’s prognosis [226]. It has been shown that micrometastases can already develop 5 years before treatment of the primary tumor [227]. Therefore, early detection of uveal melanoma is mandatory. For diagnosis, fundus examination is necessary. Suspicious choroidal lesions with documented growth, presence of subretinal fluid, and orange pigment are suggestive for uveal melanoma [224]. Further diagnostic features are ultrasonography, where a mushroom-like configuration and low internal reflectivity are typical, fluorescein angiography, indocyanine green angiography to visualize intrinsic tumor vasculature, computed tomography, and magnetic resonance imaging [220,224]. Established therapeutic options are transpupillary thermotherapy, focal radiation therapy, local resection, enucleation, or orbital exenteration [220,224]. Figure 4 shows representative pictures of patients with uveal melanoma.

A variety of clinical trials on targeting β-ARs to treat ocular hemangiomas/hemangioblastomas has been conducted. The trials are listed in Table 2.

### 4.7. β-ARs as Therapeutic Targets in Uveal Melanoma

In cutaneous melanoma, β-ARs were found to be a new target for inhibition of tumor growth and dissemination [228]. In different cell lines of cutaneous melanoma, a pro-tumorgenic effect with increased MMP-dependent motility, and increased levels of IL-6, IL-8, and VEGF by catecholamines was shown. These effects could be reversed by the non-subtype-selective β-AR antagonist propranolol [229,230]. In a retrospective study on patients with cutaneous melanoma receiving regularly β-blockers, a lower disease progression and mortality rate was found [228,231].

Immunohistochemical studies demonstrated expression of β_1_- and β_2_-ARs in all patients with uveal melanoma, and a higher expression was found in more aggressive epitheloid cells [232]. The epitheloid cell type is associated with a poorer prognosis due to a higher mitotic activity, a higher microvascular density, and more tumor-infiltrating macrophages than in the spindle cell type [233]. A possible role of the β_3_-AR in melanoma growth and vascularization was identified in cutaneous melanoma cells of mice [113]. However, the role of β_3_-ARs in uveal melanoma remains unclear. In cutaneous and uveal melanoma cell lines, potent anti-proliferative effects of propranolol have been shown in a dose-dependent manner [232]. Decreased levels of VEGF in human uveal melanoma cells were reported by propranolol treatment [232]. Expression of VEGF-A has been shown to stimulate proliferation in uveal melanoma cells [234]. Furthermore, VEGF levels are significantly higher in patients with metastatic uveal melanoma disease than in patients without metastases [235]. This proliferative effect could be inhibited by blocking VEGF-A [234]. Other authors revealed limited effects of VEGF-blockers, such as bevacizumab, on cell proliferation in uveal melanoma [236]. Likewise, cytotoxic effects of propranolol via induction of apoptosis and cell cycle arrest could be demonstrated [232]. It is known that norepinephrine activates the raf-1 kinase/MAP kinase cascade through stimulation of β-ARs [237]. Activation of the MAP kinase cascade is also involved in the development of uveal melanoma, but without involvement of the protooncogenes NRAS and BRAF, which are part of cutaneous melanomagenesis [238]. Via the MAP kinase pathway and IP3 signaling with consecutive increase in intracellular calcium, PKC is activated. PKC stimulates the RAF/MEK/ERK pathway that is associated with increased proliferation, migration, and survival in cutaneous and uveal melanoma [232,239]. By blockade of these signaling pathways, β-blockers may be beneficial in the treatment of cutaneous and uveal melanoma.

Janik et al. analyzed expression of β_2_-ARs in primary cutaneous (FM-55-P), primary uveal (92-1, Mel202), and metastatic cutaneous (A375) melanoma cells, and found cell line-dependent differences in β_2_-AR expression with higher expression levels in primary uveal melanoma cells. Furthermore, the authors could show that, through adrenaline treatment, a stimulation of melanoma cell proliferation and activation of MMPs was induced. Especially, uveal melanoma cells showed higher migration rates in comparison to cutaneous melanoma cells [240]. Patients with uveal melanoma positive for MMP-2 and MMP-9 had a significantly higher incidence of metastatic disease and lower survival rate, indicating that proteolytic enzymes, such as MMPs, may play a central role in tumor spread [241,242].

These findings indicate that blockade of β-ARs could be a potential therapeutic tool to treat uveal melanoma. However, further experimental and clinical examinations are necessary in this field. A scheme of potential effects of propranolol treatment on uveal melanoma growth is demonstrated in Figure 5.

## 5. Future Directions in Research

To expedite the clinical use of β-blockers in ocular diseases and tumor treatment, the intracellular signaling pathways, such as inflammatory, redox, and cell death signaling routes, modulated by activation or blockade of individual β-AR subtypes, remain to be characterized in more detail. Since different ocular tissues are equipped with various β-AR expression patterns, as well as with different intracellular signaling molecules, conclusions regarding individual β-AR functions cannot be generalized, but need to be confined to a specific cell type [3]. Since the immunologic system contributes to most ocular diseases, and because β-ARs are involved in various immune cell actions, expression and function of β-AR subtypes in individual immune cell types deserves further in-depth analysis [243,244,245]. Genetically modified cell cultures or animals may be useful to determine the role of individual β-AR subtypes in the pathophysiology and treatment of specific ocular diseases [107,246,247]. Since the specificity of AR antibodies is still limited, better specification of antibodies directed against-individual β-AR subtypes is needed to draw reliable conclusions regarding their expression pattern and modulation of expression by pharmacological or genetic tools [248,249,250,251]. Furthermore, more specific pharmacological agonists and antagonists for individual β-AR subtypes are desirable [252,253,254]. The use of highly selective β-AR ligands may pave the way for more specific therapeutic applications with limited side effects.

## 6. Conclusions

Disturbed β-AR signaling is discussed in the pathophysiology of some ocular surface diseases, such as dry eye disease or disturbed corneal wound healing. Moreover, β-ARs are involved in the regulation of outflow facility of the trabecular meshwork, and in the regulation of aqueous humor formation in the ciliary body. Therefore, antagonism of the β_2_-AR by timolol plays a central role in glaucoma therapy. β-ARs are also involved in new blood vessel formation and tumor growth. In different ocular types of hemangioma, such as periorbital infantile hemangioma, choroidal hemangioma, retinal hemangioblastoma, and conjunctival hemangioma, different studies on therapeutic administration of β-blockers have been introduced, reporting promising results. Furthermore, expression of β-ARs and inhibitory effects of β-blockers in uveal melanoma cells have been reported. From a clinical point of view, the specific role of each β-AR subtype in tumor growth and treatment needs to be pursued further to tailor specific therapeutic approaches.

## Figures and Tables

**Figure 1 ijms-24-04698-f001:**
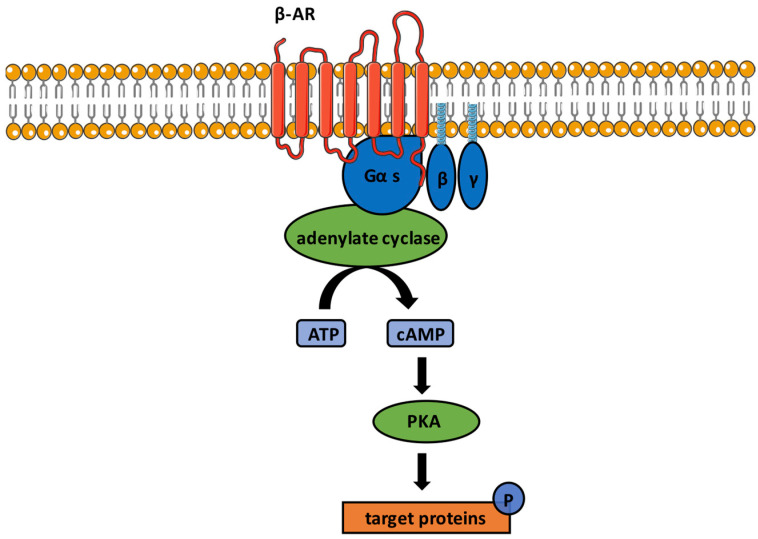
Scheme of β-adrenergic signaling pathways. Ligand binding to β-ARs leads to interaction of the receptor with coupled G protein Gαs and dissociation of Gα-GTP and Gβγ subunits. Through activation of adenylate cyclase, the second messenger cyclic adenosine monophosphate (cAMP) is produced. Via cAMP-mediated activation of proteinkinase A (PKA), target proteins, such as transcription factors or gated ion channels are phorphorylated.

**Figure 2 ijms-24-04698-f002:**
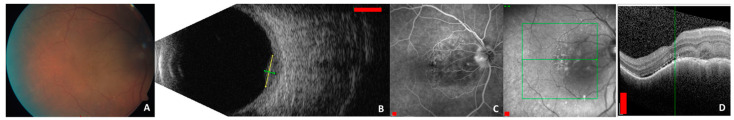
Circumscribed choroidal hemangioma in the posterior pole. Ophthalmoscopic appearance as an orange choroidal mass with margins that blend with the surrounding choroid (**A**). On B-scan ultrasonography (red scale bar is 5 mm), the hemanigoma appears as a smooth-contoured, dome-shaped choroidal mass of 1.87 mm thickness (green marking line) and horizontal extension of 6.03 mm (yellow marking line) (**B**). On fluorescein angiography (red scale bar is 400 µm), the hemangioma appears as a hyperfluorescent mass with a fine lacy network of intrinsic vessels in the choroidal filling phase (early views). The hyperfluorescence increases through most of the phases of the angiogram, with variable amounts of late leakage (**C**). Optical coherence tomography (OCT) (red scale bars are 400 µm) can be used to evaluate secondary retinal morphologic changes, such as shallow subretinal fluid or cystoid macular edema. Within the green rectangle, OCT scans of the retina have been made. The green line marks the section through the fovea (**D**).

**Figure 3 ijms-24-04698-f003:**
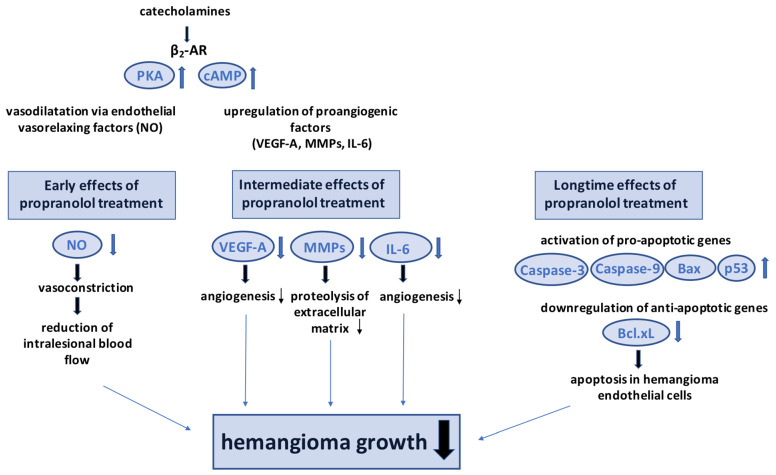
Scheme of early, intermediate and longtime effects of β-blockers on hemangioma growth. Early effects of propranolol treatment in ocular hemangioma could be due to vasoconstriction through inhibition of endothelial vasorelaxing factors, such as NO. Downregulation of proangiogenic factors, such as VEGF-A, MMPs, or IL-6, represents a possible intermediate effect of propranolol treatment. Longtime effects could be explained by induction of apoptosis in hemangioma endothelial cells.

**Figure 4 ijms-24-04698-f004:**
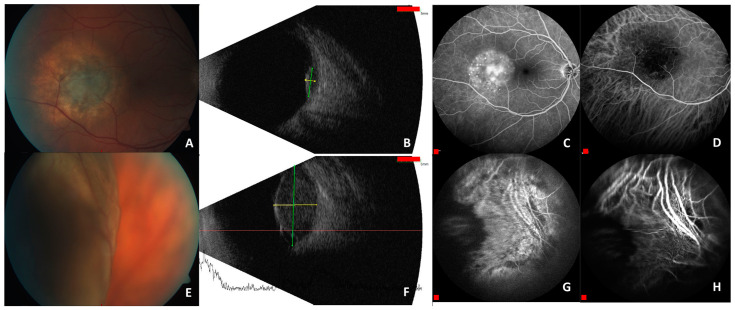
Representative pictures of uveal melanoma in a 55-year-old (**upper row**) and a 69-year-old patient (**lower row**). Ophthalmoscopic appearance of a small choroidal melanoma, located temporally to the fovea with presence of pigment epithelium alterations over the surface and confluent orange pigment marginally (**A**). B-scan ultrasound (red scale bar is 5 mm) shows small (**B**) and medium-sized (**F**) tumors that still contain an intact Bruch’s membrane and, therefore, are dome shaped. Tumor thickness was 2.02 mm (yellow marking line) with a horizontal extension of 5.99 mm (green marking line) (**B**) and 8.97 mm (yellow marking line) with a horizontal extension of 16.84 mm (green marking line) (**F**). Choroidal melanoma is almost always accompanied by secondary exudative retinal detachment, which gradually spreads from the tumor surface to the inferior periphery (**E**). Angiography with fluorescein or indocyanine green (red scale bar is 400 µm) demonstrates its value particularly for small tumors to visualize intrinsic tumor vasculature. In late sequences, fluorescein leakage with pinpoints (**C**) and choroidal vessel network in indocyanine green angiography (**D**) are typical findings. In larger tumors pigment epithelium alterations and necrotic areas of the tumor can lead to a blockade of the underlying choroidal fluorescence (**G**,**H**).

**Figure 5 ijms-24-04698-f005:**
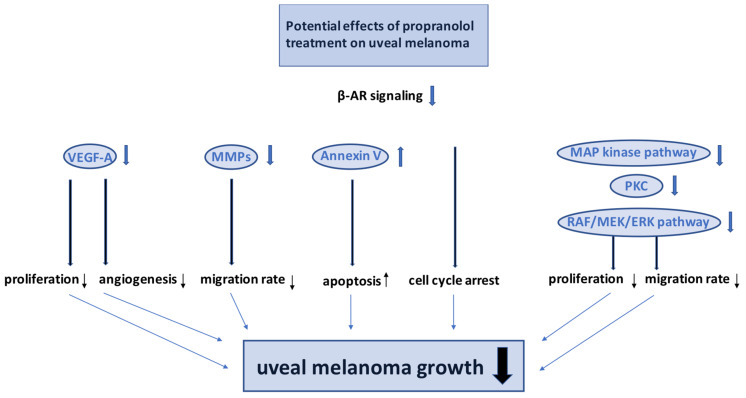
Scheme of potential effects of propranolol treatment on uveal melanoma growth. Via downregulation of pro-angiogenic and pro-tumorgenic factors, such as VEGF-A and MMPs, tumor angiogenesis, proliferation, and migration rate decreases. Annexin V represents a marker for apoptotic cells and propranolol treatment has also been shown to induce apoptosis and cell cycle arrest in uveal melanoma cells. By inhibition of MAP kinase pathway, PKC and RAF/MEK/ERK pathway proliferation, and migration rate of uveal melanoma cells is also reduced.

**Table 1 ijms-24-04698-t001:** Overview of the expression and function of β-AR in ocular structures.

Structure	β_1_-AR	β_2_-AR	β_3_-AR	Possible Functions	Relevance for Ocular Diseases	References
Cornea		x		Re-epithelialization	Corneal woundhealing	[10,11,12,13,14,15,16,17,18,19,20,21,22,23]
Conjunctiva	x	x	x	Secretion of mucous substances Mobilization of conjunctival eosinophils	Allergic conjunctivitis Dry eye disease	[24,25,26,27,28,29,30,31,32]
Lacrimal gland	x	x	x	Regulation of tear secretion Regulation of protein secretion Stimulation of lipid synthesis	Dry eye disease	[33,34,35,36,37,38,39,40,41,42]
Trabecular meshwork		x		Augmentation of outflow facility		[43,44,45,46,47,48]
Ciliary body	x	x		Aqueous humor formation	Glaucoma treatment (lowering eye pressure by nonselective β-AR blocker)	[49,50,51,52,53,54,55,56,57,58,59,60,61,62,63,64]
Choroid	x	x	x	Regulation of vascular tone Choroidal endothelial cell invasion and elongation Regulatory process of proangiogenic factors (β2-AR)	Formation of choroidal neovascularizations	[54,65,66,67,68]
Retina	x	x	x	Hypoxia mediated increase in proangiogenic factors Pericyte survival Regulation of inflammatory molecules	Retinal neovascularizations in diabetic retinopathia or retinopathia of prematurity	[17,94,95,96,97,98,99,100,101,102,103,104,105,106,107,108,109,110,111,112,113,114]

**Table 2 ijms-24-04698-t002:** Clinical trials aimed at treating ocular vascular tumors by targeting β-ARs.

Title	Disease	β-AR Antagonist	Type of Clinical Trial	Trial Registration	Outcome
Propranolol in a case series of 174 patients with complicated infantile haemangioma: indications, safety and future directions [131]	Infantile hemangioma	Oral propranolol	Prospective case series (174 patients)	Not stated in pubication	Sucessful treatment in 99.4%
Oral propranolol: an effective, safe treatment for infantile hemangiomas [132]	Infantile hemangioma	Oral propranolol	Prospective clinical study (30 patients)	Not stated in publication	Complete lesion resolution in 60%, 50% reduction in size in 20%, less than 50% reduction in size in 16.6%, resistance to treatment in 3.3%
A retrospective analysis of systemic propranolol for the treatment of complicated infantile hemangiomas. [133]	Infantile hemangioma	Oral propranolol	Retrospective study (207 patients)	Not stated in publication	Successful treatment in 99.5%
Propranolol vs. corticosteroids for infantile hemangiomas: a multicenter retrospective analysis [137]	Infantile hemangioma	Oral propranolol vs. oral corticosteroids	Multicenter retrospective analysis (110 patients)	Not stated in publication	Clearance of 75% or more in 82% of patients treated with propranolol and in 29% of patients treated with corticosteroids
A controlled study of topical 0.25% timolol maleate gel for the treatment of cutaneous infantile capillary hemangiomas [143]	Infantile hemangioma	Topical 0.25% timolol maleate gel	Retrospective, consecutive, nonrandomized, comparative single-masked cohort study (23 patients)	Not stated in publication	Good response in 61.5%, moderate response in 30.8% and poor response in 7.7% in the treated group
Efficacy and Safety of Topical Timolol for the Treatment of Infantile Hemangioma in the Early Proliferative Stage: A Randomized Clinical Trial [146]	Infantile hemangioma	Topical 0.25% timolol maleate gel	Multicenter, randomized, double-blind, placebo-controlled, phase 2a pilot clinical trial (69 patients)	NCT02731287,EudraCT Number: 2013-005199-17	No significant differences between timolol and placebo for complete or nearly complete IH resolution
Combined Oral and Topical Beta Blockers for the Treatment of Early Proliferative Superficial Periocular Infantile Capillary Hemangioma [147]	Infantile hemangioma	Oral propranolol and topical 0.25% timolol maleate gel or oral propranolol only	Randomized, controlled comparison trial (25 patients)	Not stated in publication	Hemangioma Activity Score was significantly improved in both groups, significantly better response in the systemic and topical treated group
To compare intralesional and oral propranolol for treating periorbital and eyelid capillary hemangiomas [150]	Infantile hemangioma	Oral propranolol or intralesional propranolol hydrochloride	Pilot randomized control trial (20 patients)	Clinical Trials Registry of India: CTRI/2017/08/009440	No difference in area reduction, change in appearance, ptosis and side effects between the two groups
Effects of oral propranolol for circumscribed choroidal hemangioma [175]	Choroidal hemangioma	Oral propranolol	Prospective, longitudinal interventional study (5 patients)	Not stated in publication	No clinical or diagnostic changes in tumor size during treatment
Intravitreal metoprolol for circumscribed choroidal hemangiomas: a phase I clinical trial [179]	Choroidal hemangioma	Intravitreal metoprolol	Prospective interventional phase I clinical trial (5 patients)	Not stated in publication	No signs of acute ocular toxicity
Evaluation of the safety and effectiveness of oral propranolol in patients with von Hippel-Lindau disease and retinal hemangioblastomas: phase III clinical trial [192]	Retinal hemangioblastoma in von Hippel-Lindau disease	Oral propranolol	Prospective interventional phase III clinical trial (7 patients)	EudraCT Number: 2014-003671-30.	Number and size of retinal hemangioblastomas remained stable

## Data Availability

Since no new data were created or analyzed in this study, data sharing is not applicable to this article.

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
