# Peer review of "β-Adrenoreceptors as Therapeutic Targets for Ocular Tumors and Other Eye Diseases—Historical Aspects and Nowadays Understanding"

_ijms, 2023, doi:10.3390/ijms24054698_

Round 1

Reviewer 1 Report

The current manuscript is a commendable effort from the authors to discuss and summarize present status of β-Adrenoreceptors as therapeutic targets for ocular tumors and other eye diseases. This review systematically discusses ocular tissue specific expression of different β-Adrenoreceptors and their functional roles in physiological and pathophysiological conditions, therapeutic avenues available targeting several β-Adrenoreceptors and their involvement in different ocular tumors. The tables, schematic representations and figures used in the manuscript play critical roles in terms of summarizing the current knowledge and to explain the subject better to the readers. This is a very well-written manuscript and will serve as a great resource to the ocular researchers and physicians. 

1. Authors may consider adding a comprehensive table listing the clinical trial (CT) related information citing β-Adrenoreceptors agonist/antagonist, disease state, phase of CT, clinicaltrials.gov identifier and outcome. That will be immensely helpful in improving the quality of the manuscript.

2. Rename the conclusions sections to ‘Conclusions and future directions’ and try to include more discussion on future direction based on the current knowledge in the concerned area of research and the author’s intellectual input to further our knowledge.  

Minor:

1. Page 2, lines 47-49: Delete          

Author Response

  1. Authors may consider adding a comprehensive table listing the clinical trial (CT) related information citing β-Adrenoreceptors agonist/antagonist, disease state, phase of CT, clinicaltrials.gov identifier and outcome. That will be immensely helpful in improving the quality of the manuscript.

To 1) According to the author’s suggestion, we added Table 2 in the manuscript.

  1. Rename the conclusions sections to ‘Conclusions and future directions’ and try to include more discussion on future direction based on the current knowledge in the concerned area of research and the author’s intellectual input to further our knowledge.

To 2.) We added a future direction section.

3.) Minor: Page 2, lines 47-49: Delete         

To 3.) We deleted this statement.

Reviewer 2 Report

Dear Authors,

Really nice and well organized review. However, I have some minor objections:

1) Please, include some methodology subsection, where the plan of this review is mentioned and the search is described: literature search key words, data basis used, papers inclusion/exclusion criteria, time when the literature was collected;

2) Fig. 1 and Fig. 3 are nice, but requests description about the visible picture in A, B, C, D.

3) Before the Conclusions I would like to ask the authors to include future directions in research about the topic;

4) Conclusions, sorry, are not an Abstract. Please, shorten, make them more precise and structured.

5) Nice References, thank you. But, - I noticed that there are 57 (out of 238) from the previous century. Well, after double-check I agree that they nicely fit into the text, but thus I would like to ask the authors to change slightly the title and include some historical aspects, for instance: "....your title and then the following - ... from historical aspects to nowadays understanding." (or similar to this, but with inclusion of the History"not lo lose the References!

Otherwise, - thank you very much - I enjoyed the reading of your review very much!

Author Response

  • Please, include some methodology subsection, where the plan of this review is mentioned and the search is described: literature search key words, data basis used, papers inclusion/exclusion criteria, time when the literature was collected.

To 1 According to the reviewer’s suggestion, we included a statement on literature search (lines 31 to 42).

  • 1 and Fig. 3 are nice, but requests description about the visible picture in A, B, C, D.

To 2 We made more detailed figure legends, as suggested by the Reviewer.

  • Before the Conclusions I would like to ask the authors to include future directions in research about the topic.

To 3       We included a section on „Future Directions in Research“ after the Discussion (lines 668 to 686).

  • Conclusions, sorry, are not an Abstract. Please, shorten, make them more precise and structured.

To 4       According to the Reviewer’s suggestion, we made the conclusion more concise.

  • Nice References, thank you. But, - I noticed that there are 57 (out of 238) from the previous century. Well, after double-check I agree that they nicely fit into the text, but thus I would like to ask the authors to change slightly the title and include some historical aspects, for instance: "....your title and then the following - ... from historical aspects to nowadays understanding." (or similar to this, but with inclusion of the History"not lo lose the References!

To 5       According to the Reviewer’s suggestion, we changed the title and included a subheading mentioning historical aspects.

Reviewer 3 Report

The paper entitled “β-Adrenoreceptors – Therapeutic Targets for Ocular Tumors and Other Eye Diseases” is a study based on the expression and function of adrenoreceptors in ocular structures in addition to the role of these receptors in the potential treatment of ocular diseases and tumors.

The aim of this review was to evaluate the association of these receptors with the development and progression of various tumor types and diseases. The review also provides a description of how these receptors can be considered potential therapeutic targets for ocular neoplasms, such as ocular hemangioma and uveal melanoma.

The paper is thorough and highlights the important issues behind the function and pathways of different adrenoreceptors. The use of headings and subheadings based on anatomic structures and diseases gives the paper structure and a logical organization of specific correlated topics.   

Minor editing can improve the English and flow of the text. The authors should consider a figure showing the general pathways and activation mechanisms of these receptors to summarize the details presented in the text.     

The study adds to the literature by providing an overview of pathways and mechanisms involved in different structures of the eye and in various ocular pathologies with regard to adrenoreceptors.

The study has been correctly planned and represents a solid basis for future studies regarding potential novel targets for diagnosis and treatment. It is nicely written and of clinical interest. References are appropriate. The figures and table are pertinent, and descriptive and assist in describing the results.

Author Response

  • Minor editing can improve the English and flow of the text. The authors should consider a figure showing the general pathways and activation mechanisms of these receptors to summarize the details presented in the text.

To 1       We edited the text and included a figure on the β-AR pathways.